# A Scoping Review of the Validity and Reliability of Smartphone Accelerometers When Collecting Kinematic Gait Data

**DOI:** 10.3390/s23208615

**Published:** 2023-10-20

**Authors:** Clare Strongman, Francesca Cavallerio, Matthew A. Timmis, Andrew Morrison

**Affiliations:** Cambridge Centre for Sport and Exercise Sciences, Anglia Ruskin University, East Road, Cambridge CB1 1PT, UK; francesca.cavallerio@aru.ac.uk (F.C.); matthew.timmis@aru.ac.uk (M.A.T.); andrew.morrison@aru.ac.uk (A.M.)

**Keywords:** smartphone, gait, walk, reliability, validity, review

## Abstract

The aim of this scoping review is to evaluate and summarize the existing literature that considers the validity and/or reliability of smartphone accelerometer applications when compared to ‘gold standard’ kinematic data collection (for example, motion capture). An electronic keyword search was performed on three databases to identify appropriate research. This research was then examined for details of measures and methodology and general study characteristics to identify related themes. No restrictions were placed on the date of publication, type of smartphone, or participant demographics. In total, 21 papers were reviewed to synthesize themes and approaches used and to identify future research priorities. The validity and reliability of smartphone-based accelerometry data have been assessed against motion capture, pressure walkways, and IMUs as ‘gold standard’ technology and they have been found to be accurate and reliable. This suggests that smartphone accelerometers can provide a cheap and accurate alternative to gather kinematic data, which can be used in ecologically valid environments to potentially increase diversity in research participation. However, some studies suggest that body placement may affect the accuracy of the result, and that position data correlate better than actual acceleration values, which should be considered in any future implementation of smartphone technology. Future research comparing different capture frequencies and resulting noise, and different walking surfaces, would be useful.

## 1. Introduction

As smartphone technology becomes more ubiquitous, using the sensors of the phones in our pockets becomes a cheap and convenient method to gather gait data. The use of mobile phones to evaluate human movement and diagnose and track pathological gait becomes an effective way for practitioners to gather and evaluate data, but a key concern for use in clinical practice would be the accuracy of these data. Despite the increasing use of mobile phone technology within our daily lives, the development of apps to exploit the sensors available within these devices appears more limited, which may be due to concerns about the accuracy of these data when compared to the existing methods of data collection, such as motion capture or inertial movement units used in a laboratory setting.

Whereas previous studies have reviewed wearable technology in gait more generally [1,2,3] or when wearables are used to evaluate a specific clinical pathology [4,5,6,7], it is important to remember that smartphones are simply not designed for gait analysis, unlike other wearable technology. Therefore, these devices may be considered as less accurate and more prone to error due to accelerometer data capture not being their primary use. To evaluate the accuracy of these devices in measuring kinematic data, it is important to compare smartphones to other gold-standard technology such as motion capture, force plates, or research-standard accelerometers, and evaluate the concurrent validity and/or inter-method reliability of each measure [8]. As smartphone use is so widespread, evaluating the reliability and validity of this technology allows us to conclude whether simple smartphone apps can be use in gait analysis to capture kinematic parameters, and the issues and protocols that need to be considered to ensure that these data are consistent and valuable.

This scoping review was conducted to systematically evaluate research quantifying concurrent validity and/or inter-method reliability comparing smartphone accelerometers to gold-standard measures. This will allow the identification of key themes and approaches used and the identification of any gaps in that research to inform future work in this area.

## 2. Methods

### 2.1. Protocol

This study follows the methodology for scoping reviews established in Arksey and O’Malley [9] and extended by Levac et al. [10]. In addition, the approach and execution of this review have been informed by the updated guidance issued by the Joanna Briggs Institute Scoping Review Methodology Group [11]. The preferred reporting Items for systematic reviews and meta-Analyses (PRISMA) statement extension for scoping reviews [12] has been followed to structure the reporting of this review, and a completed PRISMA-ScR checklist can be found in Appendix A.

### 2.2. Eligibility Criteria

Studies were considered eligible if they evaluated the concurrent validity or inter-method reliability of smartphone accelerometer data. There were no restrictions based on publication date; but as the search considered smartphone data, this was expected to be limited to studies since approximately 2000 due to the evolution and uptake of smartphone use. Reviews and conference papers were excluded, but these were manually checked to ensure that any relevant citations were included in the review. Papers published in languages other than English were included assuming English translations were also available.

Studies were excluded if they considered balance rather than gait parameters, or assessed static rather than dynamic movement. Further, studies were excluded unless they compared the accelerometer data (from a smartphone) with another method of objective kinematic data collection; for example, motion capture or inertial measurement units. Where studies only considered distance or time walked, such as the 6-min walk test, or total minutes of physical activity, these were excluded as no kinematic gait characteristics were evaluated. Where studies included a mixture of both gait and balance tasks, such as the timed up and go test, these were only included if the walking section of the trial was used to evaluate kinematic data such as stride time or step length. There were no restrictions placed on the operating system or type of smartphone used.

### 2.3. Information Sources

An electronic search of three databases was performed (PubMed, SportDiscus, and Web of Science) to identify relevant papers for inclusion. The search strategy was developed by three authors (C.S., M.A.T., A.M.) and refined via discussion. Google Scholar was used to check for any additional grey literature to identify unpublished studies and reduce publication bias. The final search results were exported into RefWorks. The literature search was performed between 23 and 24 September 2023.

### 2.4. Search

The search strategy included the following keywords:

(gait OR walk* OR ambul*)

AND (smartphone OR phone OR android)

AND (valid* OR reliab* OR accur*)

No further refinement or restriction was placed on the search to ensure the maximum number of studies were returned for consideration and to maximise recall.

### 2.5. Selection of Sources of Evidence

Studies were selected following abstract and keywords review and subsequent full text screening. To ensure consistency, one author (C.S.) performed the screening and applied the exclusion criteria, and this was validated by other authors (M.A.T., A.M.). Any paper considered valid for inclusion was then full-text screened and studies were included based on a consensus between all authors.

### 2.6. Data Charting

The data charting form was based on a previous scoping review conducted by this research group [13] and refined via discussion based on the scope of this review. Data charting was initially conducted in Excel (Microsoft 365, version 2309) by one author (CS) and then reviewed for accuracy (M.A.T., A.M.). Revisions to the data charting form were made iteratively via ongoing discussion as different themes emerged from the studies under review.

### 2.7. Data Items

The data extracted from each study included the demographic information for the participants and the pathological condition considered (if any). We also extracted information on the methods, including the comparator, the app name being evaluated (if provided), capture frequencies compared, the location(s) of the phone during the trials, and the nature of the trial (overground, laboratory walkway, treadmill). The duration and speeds of each trial were extracted, and the gait characteristic(s) under analysis. In addition, the method of assessing validity and/or reliability, including any sample size considerations, were extracted to allow the synthesis of approaches.

In common with other scoping reviews, an overall measurement of study quality has not been performed, but relevant study characteristics relating to methodological quality have been extracted for synthesis to gain an understanding of the development of the study protocols and the potential gaps in methodology [14,15].

### 2.8. Synthesis of Results

Studies were grouped based on the gait characteristics considered, the comparison to the type of laboratory kinematic data collected, and the method of evaluating validity and/or reliability. Any systematic reviews resulting from the search were reviewed to ensure any relevant citations were also included in the studies, as appropriate.

## 3. Results

### 3.1. Study Selection

The screening and exclusion of papers is shown in Figure 1, following PRISMA reporting guidelines [16].

After duplicates were removed, 3056 studies were considered valid for screening. A total of 2427 studies were excluded from this review as they did not evaluate kinematic data relating to gait, 141 did not use the smartphone as the primary method of data collection, and 72 used sensors other than the accelerometer (for example, video capture). In total, 72 studies did not specifically assess agreement, concurrent validity, or inter-method reliability, and 41 were excluded due to not including a comparison to a gold-standard method (for example, only evaluating the test–retest reliability of the smartphone). A total of 124 papers were excluded as these presented reviews, study protocols, and conference papers. The remaining papers were considered eligible for review.

### 3.2. Study Characteristics

The basic demographic information for each of the studies included is shown in Table 1 below, the mean and standard deviation are shown unless specified, and left blank if these values were not provided in the paper reviewed. Ages and mass have been rounded to 1 decimal place, and heights to 2 decimal places, if supplied at a higher precision, and stated as-is if provided at the lower precision. The studies are presented in reverse chronological order to show changes in reporting/methods over time.

Three studies did not include information about the biological sex of the participants [24,31,36]. Overall, the studies reviewed have recruited more females (*n* = 296) than males (*n* = 226), so this is not fully representative of the average population. Further, it is recognised that gait is affected by biological sex in both healthy adults [38] and within a pathological population [39]. As these studies are all comparing two measures when evaluating the same individual’s gait, then any difference in biological sex may not be considered important, as long as variety is represented, but this is not explicitly discussed.

The study location has been determined from the methods sections, or the author affiliations if not stated. In four studies [26,28,29,32], it has not been possible to determine the location of the data collection, with the authors being affiliated with both Thailand and the USA.

Many studies focus on healthy participants, but pathological populations are also represented, in particular Parkinson’s disease. A broad range of ages are represented in these papers, which suggests that the research conducted is generalisable to a wider sample. The mass of the participants in each sample is infrequently reported, and none of the studies had exclusion criteria relating to mass or BMI, which may suggest that the researchers do not consider this a confounding variable when assessing gait characteristics despite the potential accuracy issues due to soft tissue artefacts [40].

### 3.3. Results of Individual Sources of Evidence

Details of ‘gold standard’ comparator and smartphone information and walking protocols are presented in Table 2 and Table 3 below.

### 3.4. Synthesis of Results

#### 3.4.1. Equipment

Studies use different methods of data capture for the comparator technology, but twelve studies use motion capture to determine changes in marker position. Some studies also include additional technology such as footswitches [17], IMU [25,31,36] or video [29]. Other equipment types used as the comparator were based on IMUs [24,27,30], accelerometers [37], or pressure-sensitive walkways [21,22,32,34], and one study captured video and identified gait events from this for comparison with the smartphone data [26].

#### 3.4.2. Capture Frequency

The capture frequencies used for the smartphones varied from 15 Hz to 100 Hz. Four studies [21,29,32,37] document using the Android SENSOR_DELAY_FASTEST setting [41], which uses the fastest possible available capture rate, which has increased over time as smartphone technology has improved.

The capture frequencies for the comparator are often matched to the smartphone capture frequency or set to a larger value and then resampled to the same time points.

#### 3.4.3. Location of Markers and Phone

The number of markers used with the motion capture technology varied from a single marker to a full-body 53 marker set. Markers were often placed on or near the smartphone [18,23,36,37]. In seven studies, the smartphone was placed in an appropriate place that would replicate day-to-day use, for example, a front pocket [17,24,25,26,27], or in different locations to evaluate whether a change in body position affected the reliability [28,29,32]. Placement of the smartphone on the lumbar spine was also used [18,19,21,23,29,30,33,35,36,37], as this is often used as the standard placement for accelerometers to evaluate movement and determine lower body gait events [42]. In addition, one study placed the smartphone on the sternum [31] and one on the navel [34].

#### 3.4.4. Walking Protocols

Studies mostly use preferred walking speed, although the protocol for determining these speeds is often lacking in the method description. The protocol for determining the preferred walking speed is stated explicitly in two studies [17,20] and the cues used to initiate the participants are stated in two studies [24,29]. Some studies vary the speed to evaluate if this affects the accuracy of comparison between the smartphone and ‘gold standard’ device—in one study [20] a fixed speed is used which is specified numerically, one uses metronome cueing to fix the average speed and increases this by 10% [34], another [24] specifies the verbal cues used to obtain a fast or slow speed, whereas other studies that consider speed changes do not clearly explain the protocol to determine this [29,31,32,33].

The majority of the studies were conducted indoors, with one study [29] also using an outdoor level pedestrian walkway, and a further study considering outdoor walking and obstacle crossing [28]. Two studies used a treadmill due to the need to control the data captured or to fix speeds [17,20], and one used corridors [27], but the majority of the other studies used laboratory-based hard floor walkways [19,21,22,24,25,26,29,30,31,32,33,34,35,36,37]. One study also used the participants’ indoor home environment in addition to the treadmill [20]. The surface used in the trials was not reported in two studies [18,23].

Dual task trials are included in six studies [18,20,23,24,27,30]. Different protocols are used, with some studies dual task consisting of the participants turning their head from side to side while walking [18,23], or a cognitive task such as the ‘serial seven’ or ‘serial threes’ test [20,24,27], or a combination of both numerical and verbal cognitive tasks [30].

The duration of each trial varies, and is expressed in either distance or walking time. As one trial considers a non-linear analysis of the data [17], this requires a longer time series to fully capture the nature of the temporal gait changes and should exceed 500 stride intervals for fractal analysis [43] or 200 strides for entropy analysis [44]. The remaining studies consider linear measures such as means and coefficient of variation, and so do not have the same requirement for a long time series, and these vary from 6 steps [19] or 6 s [18,23] to 120 s of walking data [24,26]. A justification of trial length in the studies concerning linear measures has not been included in any of the papers reviewed.

Turns are included in trials in five studies [24,26,29,30,33], and are included in the trial but excluded from the subsequent analysis in four studies [20,22,27,34]. Subjects walked barefoot in five studies [18,22,25,32,35], without shoes in one study [34] and in normal shoes in five studies [17,24,27,29,37], otherwise this was not stated. Obstacle crossing and uneven surfaces were considered in one study [28]. Inclines and steps have not been included.

#### 3.4.5. Analysis

The signal processing, analysis, gait events identified, and reliability measures are summarised in Table 4 below.

Sample size calculations are explicitly included in five studies [17,18,20,24,26], and one further study states the calculated sample size but not the values or methods used to obtain it [34]. Studies that evaluate the required sample size either base the calculation on attaining an intraclass correlation coefficient (ICC) of ≥0.8 [18,20], based on the results of previous studies [17,26], or one study [24] uses the recommendations from Bujang and Baharum [45].

**Table 4 sensors-23-08615-t004:** Processing and analysis.

Study	Filtered	Resampled	Sample Size Calculation	Gait Characteristics	Determination of Characteristic	Reliability/Validity Measure
Di Bacco et al. (2023) [17]	For linear analysis only	Y	Y	Stride time DFA Entropy	As [46]	ICC B/A
Olson et al. (2023) [18]	N	N	Y	Step length Step time Periodicity	As [23]	ICC B/A
Grouios et al. (2022) [19]	N	N	N	Raw acceleration	N/A	ICC Pearson
Christensen et al. (2022) [20]	N	N	Y	Stance time Step length Cadence Stride length Swing time	Identified by researcher	ICC B/A
Kelly et al. (2022) [21]	Y	Y	N	Cadence	Positive peaks from the AP direction were identified as heel strikes	Pearson
Shema-Shiratzky et al., 2022) [22]	N	N	N	Step length Cadence Single/double support %		PearsonB/A
Rashid et al. (2021) [23]	N	N	N	Step length Step time Periodicity	A wavelet-based step-event detection algorithm and a double-pendulum gait model	ICC B/A Pearson
Shahar et al. (2021) [24]	N	N	Y	Cadence Step length Gait stance phase % Swing phase %	Not stated	ICC B/A
Alberto et al. (2021) [25]	Y	Y	N	Stride duration Stance phase duration Stride length Cadence	As [46]	B/A
Lugade et al. (2021) [26]	Y	N	Y	Step time Cadence	Video-based concurrently with accelerometer capture	B/A Pearson
Su et al. (2021) [27]	Y	N	N	Stride time Stride time variability	As [46]	Pearson
Silsupadol et al. (2020) [29]	Y	N	N	Step time Step length Cadence	Positive peaks in the filtered AP direction were identified as heel strikes	B/A Pearson
Howell et al. (2020) [30]	Y	N	N	Stride length Cadence	Positive peaks in the filtered AP direction were identified as heel strikes	ICC Pearson
Kuntapun et al. (2020) [28]	Y	N	N	Step time Step length Cadence COM displacement	Positive peaks in the filtered AP direction were identified as heel strikes COM identified via double integration of the acceleration time series	Pearson B/A
Tchelet et al. (2019) [31]	N	Y	N	Step length Cadence		B/A
Silsupadol et al. (2017) [32]	Y	Y	N	Step length Step time Cadence	Positive peaks in the filtered AP direction were identified as heel strikes	ICC B/A
Pepa et al. (2017) [33]	N	N	Y	Step period Step length	Various algorithms to identify heel strike compared	B/A Pearson
Ellis et al. (2015) [34]	N	Y	Y	Step time Step length	Peaks in AP signal	ANOVA and effect sizes
Furrer et al. (2015) [35]	Y	N	N	Step length. COM displacement.	Double integration of accelerations	B/A Pearson
Steins et al. (2014) [36]	Y	Y	N	COM position. COM acceleration.	Integration of acceleration	ICC B/A
Nishiguchi et al. (2012) [37]	Y	Y	N	Peak frequency	Peak frequency calculated from smoothed acceleration data	Pearson

Notes: ICC = intraclass correlation coefficient; B/A = Bland Altman limits of agreement; AP = anterior-posterior.

One study [17] uses non-linear analysis when evaluating reliability, specifically detrended fluctuation analysis, approximate entropy, and sample entropy of a time series without filtering/smoothing. When linear measures are considered in the same study, the data are filtered prior to analysis. In other studies that include filtering, the cut off frequencies range from 2 Hz to 20 Hz, with some studies [21,28,30,32] also adding additional filtering of the anterio-posterior signal based on previous work by Zijlstra and Hof [47].

The actual acceleration values are used in the reliability analysis in two studies [19,36], whereas the majority of the other papers consider discrete events that can be derived from the original time series (e.g., stride time).

The majority of studies included in this review use ICCs to evaluate inter-method reliability, and also include Bland Altman limits of agreement or Pearson correlation coefficients to evaluate concurrent validity in addition to this. However, when interpreting the ICC value, different ranges have been used to quantify the result. The majority of papers reviewed that implement ICCs [17,19,20,24] use the ranges specified by Koo and Li [8]; that is, <0.5 poor, 0.5–0.75 moderate, 0.75–0.90 good, >0.90 excellent. However, two papers [18,23] use ranges specified by Munro [48]: <0.50 poor, 0.50–0.69 moderate, 0.70–0.89 high, >0.90 excellent; two studies [28,32] use ranges recommended by Cicchetti [49], <0.40 poor, 0.40–0.60 fair, 0.60–0.75 good, >0.75 excellent; one study [36] uses ranges recommended by Shrout and Fleiss [50]: <0.40 poor, 0.40–0.75 fair to good, >0.75 excellent; and one study [30] uses an uncited set of ranges: ≤0.59 low, 0.60–0.69 marginal, 0.70–0.79 adequate, 0.80–0.89 high, >0.90 very high. The discrepancy between these ranges is shown in Figure 2 below.

#### 3.4.6. Findings

Many papers reported an excellent correlation either via the ICC [17], Pearson correlation coefficient [26,27,28,33,37], or Bland Altman limits of agreement [25,31]. Olson et al. [18] concluded that step time had an excellent reliability, whereas step length was good. Other papers achieved good to excellent reliability [24,29,30,35]. Kuntapun et al. [28] evaluated both level walking, irregular, and obstacle crossing, and found high to very high correlations for gait characteristics but low to high correlations for the COM displacement. Steins et al. [36] found the position data to be excellent, but the actual acceleration to only be good (>0.54). Grouios et al. [19] conclude that smartphones are a valid and reliable alternative to motion capture technology, but their results include ICC values from −0.348 to 0.796 and Pearson correlation coefficients of −0.464 to 0.460 which do not seem to support this.

Shema-Shiratzky et al. [22] evaluated both left and right sides and concluded that smartphones have an excellent validity compared with a pressure-sensitive walkway for cadence but only achieved an adequate correlation for single limb support, double limb support, and stance phase. Kelly et al. [21] also found a strong correlation between the smartphone and the walkway for cadence. When considering different body positions, Silsupadol et al. [32] found phone placement may be important, with body and belt placement resulting in an excellent reliability when compared to the gold standard, whereas bag, hand, and pocket are good.

## 4. Discussion

### 4.1. Summary of Evidence

The choice of the gold standard equipment to use to evaluate the validity and reliability of the smartphone data capture is not justified in any of the studies, so this may relate to convenience or previous studies conducted by the research groups. In particular, there are research groups and co-authors common in several papers, which may suggest that later papers develop earlier research, which could imply methodological bias. However, this also means that limitations identified in earlier papers can be further developed in later research studies, such as the lack of turns identified in the protocol for Silsupadol et al. [32], which is addressed in the 2020 paper [29].

The choice of capture frequency is important to ensure that the quickest system changes are captured, with 24 Hz suggested as the minimum for walking trials [51] due to the Nyquist sampling theorem. One study has a sampling rate (15 Hz) that may not capture all the required data [19], although low sampling rates (12.5 Hz) have been used successfully to capture data about cadence in older people with osteoarthritis [52]. However, high sampling frequencies may increase the chance of noise in the data, so clear justification of the choice of sampling frequency is needed to reduce the risk of oversampling and associated error, which may affect the evaluation of reliability if there is error present in one sample and not the other.

There is a range of different-length trials present in the reviewed papers, but this is not justified other than when discussing non-linear analysis and the requirement for many data points [17]. The trial length should also be considered in conjunction with the capture frequency to establish the number of data points available for analysis in each case—this varies in the studies reviewed from approximately 600 data points [18,23] to 12,000 captured data points [24,26] which is a considerable difference. As some of the studies include an older or pathological population, the trial length should be considered further to ensure that fatigue does not affect the gait pattern or increase the risk of adverse events.

The protocol for determining preferred walking speed is often missing from method descriptions, and this has been found to be problematic, with speed being a potential confounding variable in gait analysis with recommendations that this should be standardised to avoid ambiguity [53]. In particular, the use of specific cues can affect the speed selected by the participant [54] and result in a preferred speed that is not optimal. The protocol for choosing a self-selected speed has been specified in two studies [17,20], and the cue used in another [24], and this is important to ensure that studies are repeatable and methods are rigorously reported.

#### 4.1.1. Ecological Validity

Many studies attempt to replicate laboratory-based testing when deciding the placement of the smartphone, such as placing it strapped to the lower back or sternum. While this makes sense in terms of being a robust way of checking reliability versus gold standard technology, which may be applied in the same area, this does imply a lack of ecological validity, as this is not where research participants will be carrying their smartphone in a real-world situation. The placement of the smartphone during testing has taken this into account, with more focus on actual body positions that the smartphone may be used, such as the front pocket, or close to one hip. Further studies [28,32] have validated different body positions for the smartphone which may be used in recommendations for research participants in terms of where to keep their device during walking trials to maximise accuracy. There is limited research on smartphone location while walking, but a study of younger women (aged 15–40 years) found that the preferred smartphone locations also included hanging around the neck, or tucked into their bra [55], so further analysis on smartphone body locations and the effect of these on the reliability of kinematic data is warranted.

Similarly, walking barefoot in some trials lacks ecological validity if smartphone accelerometry data are to be used in a real-world setting. The location of the trials conducted in the reviewed studies often used laboratory walkways, with only two studies using an outdoor setting [28,29], which would replicate a real-world data collection. Various studies included in this review also included dual task components to replicate real-world data collection; however, these often involve cognitive or motor tasks that do not replicate what the participant may experience when walking in real life. Thus, rather than simply walking and talking, the dual task components include mathematical tasks or head-turning tasks, which are perhaps unrealistic. The studies reviewed suggest that dual tasking when captured via smartphone or gold standard is comparable, accurate, and reliable, which would also suggest that simpler dual task components may also have good reliability.

Turns are not dealt with consistently in the studies reviewed, with some deliberately excluding these as they disrupt stride timing [46]. In other studies, turns are included as these represent real-world gait more accurately due to the quantity of turns experienced in activities of daily living [56] and can be accurately identified within a time series [57]. As the papers reviewed are considering validity and reliability of smartphones when compared to gold standard systems, it could be argued that turns should be included as representative of usual gait, and that the two systems should handle these in the same way if we were to conclude that the smartphone was a reliable alternative measure. It should also be considered that some of the studies reviewed focused on Parkinson’s disease or older adult fallers, and turns are considered to be a contributory factor in negative events such as freezing of gait [58] or increased falls risk [59], so capturing kinematic data during turning may be particularly useful in these populations.

#### 4.1.2. Analysis

The raw acceleration data are often resampled, as smartphones do not sample at reliable time intervals and so need to be interpolated to ensure that the data points represent the same capture point. Many studies reviewed have reported the need to resample or interpolate the data, and this could be a potential cause of poor results if studies did not deal with this issue, as this would introduce lags into the time series. Various algorithms have been used to determine specific gait events, but the need to identify specific gait events rather than consistent features in the time signal has not been clearly explained. For evaluating stride time, for example, looking at peaks/troughs in the signal as the same consistent point, even though these may not correspond to a specific gait event, could be potentially as valid as identifying heel strikes to calculate this value, which has been employed as a strategy in some of the studies reviewed.

It should be noted that the Grouios et al. [19] paper attempts to test the reliability of each acceleration value gathered, whereas most other papers reviewed reduce the sample data points by extracting discrete data such as stride length to use in their reliability analysis. Steins et al. [36] also consider acceleration data directly and find that the actual raw accelerations have a fair to excellent reliability, whereas the position data obtained by double integration of the acceleration series had a higher reliability. This suggests that the analysis of data derived from discrete gait events, such as stride length or step time, may be more valid than using the accelerations more directly, suggesting that the accelerations may include more noise and potential error in the signal.

Sample size calculations are included in later studies, which may relate to increasing rigour in reporting over time with published articles having more defined reporting standards to adhere to [60]. The wide range of ranges used to determine whether reliability is ‘good’ or ‘excellent’ is not consistent in the studies reviewed, but most studies also report the numerical value of the ICC to allow comparison between studies.

There are a ranges of approaches adopted in the studies reviewed, with agreement analysed via Bland Altman, concurrent validity analysed via correlation, and inter-method reliability analysed using ICC. In some cases, the language used could be more precise to explain the choices to assess concurrent validity rather than inter-method reliability, for example, rather than more ambiguous terms such as ‘feasibility’ and ‘accuracy’. When studies use Bland Altman plots or Pearson correlations rather than ICC, this is often not justified, and one study uses an analysis of variance (ANOVA) which is much more limited in use than ICC for determining reliability [61]. Pearson correlations alone may be misleading, as these do not measure reliability or agreement between methods [62], which may be why several studies considered multiple methods of determining validity and/or reliability.

### 4.2. Limitations

A scoping review approach has been used here to evaluate the breadth and depth of research in a specific area, and to identify the approaches used to inform future research. Although we searched grey literature, it is possible that publication bias may have affected the studies included in this review. In particular, pilot or preliminary studies may not have been published in peer-reviewed journals due to small sample sizes or lack of significance [63]. As is standard with scoping reviews, an evaluation of the quality of each study has not been performed [14,15], but we have extracted key themes and approaches to allow readers to assess their methodological quality and rigour.

## 5. Conclusions

A range of different smartphone makes and models have been considered in the studies reviewed, as have differing speeds and dual task components. The reliability of smartphone-based accelerometry data has been assessed against motion capture, pressure walkways, and IMUs as ‘gold standard’ technology and has been found to be accurate and reliable. A range of different methods have been used to identify gait events, to process and analyse the data, and to evaluate the reliability. This suggests that smartphone accelerometers can provide a cheap and accurate alternative to gather kinematic data, which can be used in ecologically valid environments to potentially increase diversity in research participation.

### Recommendations for Future Research

The studies reviewed cover a range of capture frequencies but no study explicitly compared different capture frequencies to see if this affects the reliability. As smartphones are not designed to capture accelerometry data for gait analysis, then it is feasible that increasing capture frequency could add noise to the signal; thus, it would be important to consider the optimal capture frequency for smartphone use, rather than just try and capture the maximum frequency possible. In addition, a consideration of different walking surfaces would increase the generalisability of the research and how this relates to the data collection in the real world and dissemination of smartphone-based data capture ‘in the wild’.

## Figures and Tables

**Figure 1 sensors-23-08615-f001:**
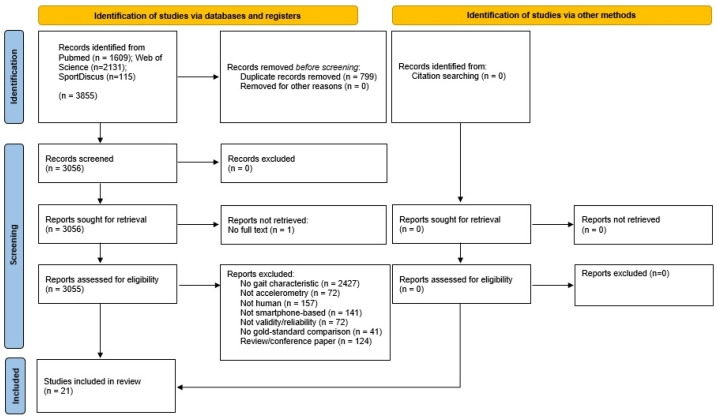
PRISMA diagram of study selection.

**Figure 2 sensors-23-08615-f002:**
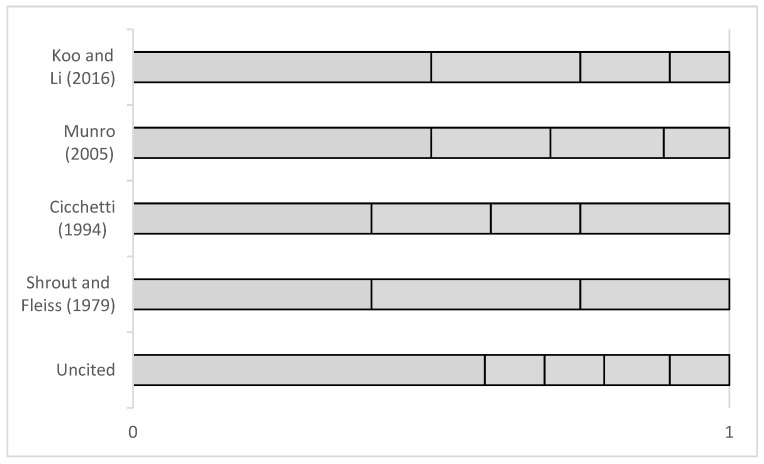
Ranges used when classifying ICC, ranging from ‘poor’/’low’ to ‘excellent’/’very high’ [8,48,49,50].

**Table 1 sensors-23-08615-t001:** Basic sample characteristics.

Study	Journal	Location	Participants	Age (Years)	Height (m)	Mass (kg)	BMI (kg·m^−2^)
Di Bacco et al. (2023) [17]	J. Biomech.	Canada	9M 8F	24.7 ± 3.7	1.73 ± 0.1	73.1 ± 14.2	
Olson et al. (2023) [18]	Gait Posture	New Zealand	14M 20F	42–92			25.3 (median)
Grouios et al. (2022) [19]	Sensors	Greece	1M	29	1.78	72	
Christensen et al. (2022) [20]	J. Orthop. Surg. Res.	USA	8M 12F healthy; 7M 5F TKA/THA	42.3 ± 19.7 58.7 ± 6.5	1.63 ± 0.24	77.0 ± 17.4	
Kelly et al. (2022) [21]	Measurement	USA	10M 13F	21 ± 2		90.0 ± 15.5	
Shema-Shiratzky et al. (2022) [22]	Gait Posture	Israel	35M 37F Knee OA (49) Ankle/hip OA (11) Low back pain (12)	57.2 ± 1.9			
Rashid et al. (2021) [23]	Sensors	New Zealand	5M 15F	46 ± 27	1.67 ± 0.17	76 ± 19	
Shahar et al. (2021) [24]	Sensors	Israel	60	37.2 ± 13.4	1.71 ± 0.10		
Alberto et al. (2021) [25]	BMC Neurol.	Portugal	12M 7F PD	62 ± 12.3			
Lugade et al. (2021) [26]	J. Aging Phy. Act.		8M 13F 7M 14F non-faller older 3M 18F faller older	22.9 ± 2.2 71.8 ± 4.5 72.9 ± 5.3	1.64 ± 0.08 1.56 ± 0.07 1.56 ± 0.07	56.1 ± 9.1 57.6 ± 5.5 56.7 ± 7.5	
Su et al. (2021) [27]	JMIR Mhealth Uhealth	China	33M 19F PD	63 ± 10	1.7 ± 0.9	70 ± 21	
Kuntapun et al. (2020) [28]	Frontiers in Sports and Active Living		3M 9F young 3M 9F older	23.4 ± 2.2 75.6 ± 5.6	1.63 ± 0.07 1.60 ± 0.09	58.3 ± 9.9 58.0 ± 6.6	
Silsupadol et al. (2020) [29]	IEEE J. Biomed.		4M 8F young 0M 12F older	21.4 ± 1.2 72.4 ± 6.1			
Howell et al. (2020) [30]	Phys. Sportsmed	USA	6M 14F	22.2 ± 2.1	1.70 ± 0.08		
Tchelet et al. (2019) [31]	Sensors	Israel	4	33.5 ± 3.9			
Silsupadol et al. (2017) [32]	Gait Posture		1M 11F younger 7M 15F older	22.7 ± 0.9 73.9 ± 5.6			21.2 ± 4.1 23.7 ± 3.6
Pepa et al. (2017) [33]	Gait Posture	Italy	8M 3F	22–30			
Ellis et al. (2015) [34]	PLoS One	Singapore	7M 5F PD 8M 4F controls	65.0 ± 8.4 63.1 ± 7.8			
Furrer et al. (2015) [35]	Gait Posture	Switzerland	10M 12F	27.4 ± 3.9	1.74 ± 0.08	65.5 ± 10.2	
Steins et al. (2014) [36]	J. Biomech.	UK	10	25.6 ± 3.5	1.73 ± 0.17	73.0 ± 17.1	
Nishiguchi et al. (2012) [37]	Telemed. J. E.Health	Japan	17M 13F	20.9 ± 2.1	1.67 ± 0.08	60.4 ± 7.7	

Notes: PD = Parkinson’s disease; TKA = total knee arthroscopy; THA = total hip arthroscopy; OA = osteoarthritis.

**Table 2 sensors-23-08615-t002:** Results from individual sources of evidence—equipment.

Study	Comparator			Smartphone		
	Equipment	Markers	SF	App/Phone (OS)	SF	Location
Di Bacco et al. (2023) [17]	Motion capture (7 camera Vicon)	Heel of right shoe.	100	- Google (Android)	100	Front right pocket
	Delsys footswitch sensor	Right heel	296			
Olson et al. (2023) [18]	Motion capture (12 camera Qualisys)	Marker in centre of phone screen, plus posterior calcaneus and head of the fifth metatarsal bilaterally		Gait&Balance iPhone (iOS)		L5/S1
Grouios et al. (2022) [19]	Motion capture (10 camera Vicon)	16 markers, lower body.	15	Accelerometer iPhone (iOS) Accelerometer Acceleration Log Samsung/Huawei (Android)	15	Lumbar spine
Christensen et al. (2022) [20]	Motion capture (10 camera Vicon)	53 markers.	200	OneStep iPhone (iOS)	100	2 phones, anterior thigh.
Kelly et al. (2022) [21]	Tekscan Strideway pressure sensitive walkway		30	Gait Analyzer LGK40 (Android)	95–105	L5
Shema-Shiratzky et al. (2022) [22]	Protokinetics Zeno pressure sensitive walkway			OneStep Samsung (Android)	100	Upper left and right thigh.
Rashid et al. (2021) [23]	Motion capture (7 camera Vicon)	One marker on the centre of the smartphone, and two were placed on each foot, at the posterior calcaneus and lateral fifth metatarsal.	200	Gait&Balance iPhone (iOS)	100	L5/S1
Shahar et al. (2021) [24]	APDM mobility lab	3 IMUs, on both feet and L5	128	OneStep (Android)	100	Front pocket
Alberto et al. (2021) [25]	Motion capture (10 camera Qualisys)	48 markers, plus clusters.	120	Kinetikos Nokia (Android)	100	Both sides front pocket
	15 × Xsens IMU	Head, thorax, scapulae, upper arms, forearms, hands, sacrum, thighs, shanks, and feet.	120			
Lugade et al. (2021) [26]	Video (gait events identified)		30	Gait Analyzer (Android)	50	Right hip
Su et al. (2021) [27]	APDM mobility lab	3 IMUs, on both feet and L5	100	- iPhone (iOS)	100	Front pocket
Kuntapun et al. (2020) [28]	Motion capture (9 camera BTS)	28 markers	120	Gait Analyzer Samsung (Android)	50	L3, bag
Silsupadol et al. (2020) [29]	Motion capture (9 camera BTS)	28 markers.	120	SensorData Samsung and Asus (Android)	100	L3, L5, bag
	Video (gait events identified)					
Howell et al. (2020) [30]	3 × Opal IMU	Feet and lumbosacral junction.	128	Gait Analyzer Samsung (Android)	50	Lumbar spine
Tchelet et al. (2019) [31]	Motion capture (10 camera Qualisys)	8 markers (shoulders, sternum, back, inside/outside feet).		Enchephalog Android and iPhone (iOS)		Sternum
	1 × Opal IMU	Sternum	128			
Silsupadol et al. (2017) [32]	GAITrite pressure sensitive walkway		80	SensorData vivo (Android)	95–105	L3, bag near right hip, front pocket (both vertical and horizontal orientation), handheld (as if speaking)
Pepa et al. (2017) [33]	Motion capture (6 cameras BTS)	9 markers on ASISx2, mid PSIS, heel, 1st, 5th metatarsal.	100	AccOrient iPhone (iOS)	100	L3. Lateral pelvis.
Ellis et al. (2015) [34]	Footswitch, sensor mat, GAITrite pressure sensitive walkway	Footswitch on heel pad.		SmartMove iPod Touch (iOS)	100	Navel
Furrer et al. (2015) [35]	Motion capture (8 camera Vicon)	34 markers.	200	- Android	50	L3
Steins et al. (2014) [36]	Motion capture (6 camera Qualisys)	L3	100	- iPod Touch (iOS)	100	L3
	1 × Xsens IMU	L3	100			
Nishiguchi et al. (2012) [37]	1 × WAA-006 accelerometer	L3	33.3	- Android	33.3	L3

Notes: SF = sample frequency, IMU = inertial measurement unit.

**Table 3 sensors-23-08615-t003:** Results from individual sources of evidence—walking protocols.

Study	Environment	Speed	Duration
Di Bacco et al. (2023) [17]	Treadmill	PWS	3 × 8 min
Olson et al. (2023) [18]		PWS, PWS + dual task	4 × 6 s
Grouios et al. (2022) [19]	6 m walkway	PWS	9 × 6 steps
Christensen et al. (2022) [20]	Treadmill, indoor home environment	Treadmill: PWS, 0.8 ms^−2^, 2 ms^−2^, PWS + dual task	Treadmill: 15 steps Home: 30 s.
Kelly et al. (2022) [21]	10 m walkway	PWS	6 × 20 m
Shema-Shiratzky et al. (2022) [22]	10 m walkway	PWS	4 × 10 m
Rashid et al. (2021) [23]		PWS, PWS + dual task	4 × 6 s
Shahar et al. (2021) [24]	10 m walkway	PWS, ‘as fast as you can’, ‘as if the floor was slippery’, PWS + dual task	2 min
Alberto et al. (2021) [25]	Walkway	PWS	3 × 10 m
Lugade et al. (2021) [26]	Lab overground, circular	PWS	2 × 2 min
Su et al. (2021) [27]	10 m hallway (turns removed in analysis)	PWS, PWS + dual task	2 × 20 m
Kuntapun et al. (2020) [28]	Walkway Outdoor area.	PWS. Indoors and outdoors, level, irregular, obstacle crossing	10 m
Silsupadol et al. (2020) [29]	Walkway Outdoor area.	Speed changes and turns in separate trials. Slow = ‘as slow as they can’ Fast = ‘as fast as they can without running’	10 m
Howell et al. (2020) [30]	Walkway	PWS. Turns included. Dual task.	5 min, 5 × 20 m (with turn).
Tchelet et al. (2019) [31]	Walkway	Various—not specified what.	3 m/5 m
Silsupadol et al. (2017) [32]	Walkway	PWS, slow, fast (actual values not specified)	10 m
Pepa et al. (2017) [33]	Walkway	PWS, higher, lower (actual values not specified)	10 m platform. Back and forth.
Ellis et al. (2015) [34]	Walkway	PWS, cued PWS, cued PWS + 10%	26 m path, turn halfway
Furrer et al. (2015) [35]	Walkway	PWS	10 × 10 m
Steins et al. (2014) [36]	Walkway	PWS	4 × 10 m
Nishiguchi et al. (2012) [37]	Walkway	PWS	3 × 20 m

Notes: PWS = preferred walking speed.

## Data Availability

No new data were created or analyzed in this study. Data sharing is not applicable to this article.

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
