# Peer review of "A Scoping Review of the Validity and Reliability of Smartphone Accelerometers When Collecting Kinematic Gait Data"

_sensors, 2023, doi:10.3390/s23208615_

Round 1

Reviewer 1 Report

Manuscript: A scoping review of the validity and reliability of smartphone accelerometers when collecting kinematics gait data

 General comments:

Overall, this scoping review was interesting to read. The scoping review process was sound and I have only minor comments surrounding the use of validity and reliability terminology.

Methods:

Page 2, line 64:  ‘Studies were considered eligible if they evaluated the validity or reliability….’ This sentence implies that ‘either’ validity of reliability research would be eligible, however, in line 73 the authors state ‘…studies were excluded unless they compared the accelerometer data (from a smartphone) with another method or objective kinematics data collection. This implies that only validity studies would be eligible. The authors need to clearly define what is meant by validity and reliability here. Validity would be the comparison between technology and reliability (as I would use it) is the test-retest reliability (different sessions) or inter-rater reliability. Page 3 Line 115 uses ‘validity and/or reliability’, same for line 123.

Page 3, line 105 – Detailed reference for Excel should be used.

Results:

Page 4: Table 1. Units for Age, Height, Mass and BMI should be added to the table titles.

Page 11, line 137: ‘two minutes’ is used. But in line 236 ‘6 seconds’ is used. Consistency is needed.

Page 13, line 255: ICC is defined, and abbreviation used but then on page 14 line 268 the abbreviation is not used.

Discussion:

Page 15, line 307: The first sentence highlights ‘reliability’ but not validity?

Page 16, line 354: ‘aged 15-40’ *unit of years should be added.

Author Response

Many thanks for your comments, your time considering this paper is appreciated.  We have corrected the omissions you identified in your comments and added clarification where requested.

Page 2, line 64:  ‘Studies were considered eligible if they evaluated the validity or reliability….’ This sentence implies that ‘either’ validity of reliability research would be eligible, however, in line 73 the authors state ‘…studies were excluded unless they compared the accelerometer data (from a smartphone) with another method or objective kinematics data collection. This implies that only validity studies would be eligible. The authors need to clearly define what is meant by validity and reliability here. Validity would be the comparison between technology and reliability (as I would use it) is the test-retest reliability (different sessions) or inter-rater reliability. Page 3 Line 115 uses ‘validity and/or reliability’, same for line 123.

Thank you for your comment, this is something that the authors discussed in the preparation of the manuscript due to the reviewed studies not being consistent in their own use of this terminology.  This review considers concurrent validity and inter-method reliability, which is explored further in the discussion section to show the differences and limitations.  Concurrent validity is usually a measure of correlation, so lacks detailed information on accuracy, whereas inter-method reliability considers accuracy (usually) via calculation of ICC so we considered it important to include both methods.  Many of the papers included specifically say they evaluate one of these or the other, so we have included both to ensure that this is representative.  We have not included papers that only include test-retest reliability for different sessions without comparison to a gold-standard.  We have revised the paper to make this clearer.

Page 3, line 105 – Detailed reference for Excel should be used.

Corrected.

Results:

Page 4: Table 1. Units for Age, Height, Mass and BMI should be added to the table titles.

Apologies for this omission, this has now been added.

Page 11, line 237: ‘two minutes’ is used. But in line 236 ‘6 seconds’ is used. Consistency is needed.

Corrected.

Page 13, line 255: ICC is defined, and abbreviation used but then on page 14 line 268 the abbreviation is not used.

Corrected.

Discussion:

Page 15, line 307: The first sentence highlights ‘reliability’ but not validity?

Corrected.

Page 16, line 354: ‘aged 15-40’ *unit of years should be added.

Corrected.

Reviewer 2 Report

This paper focuses on evaluating and summarizing existing literature that considers the validity and/or reliability of smartphone accelerometer applications when compared to “gold standard” kinematic data collection (for example motion capture). According to the authors, data provided by smartphones has been found to be accurate and reliable, so smartphone accelerometers can provide a cheap and accurate alternative to acquire kinematic data.

The authors are suggested to focus on the following points:

1.- References should be cited as [1-3] instead of [1,2,3]. Your reference manager should allow this format.

2.- The Introduction section must be redone or complemented. It must include the purpose, importance and need of this study and the novelties and contributions with respect the state of the art.

3.- The authors devote a lot of efforts in explaining the screening methodology used but finally only 21 papers were selected for the core of the scooping review. Review paper usually are based on over 100 papers/documents.

4.- From my point of view this study misses important points such as a scientific discussion of the minimum video frame rate according to each specific application (speed race, long jump,…), positioning accuracy (the positioning accuracy of GPS or Wi-Fi systems, typically some few meters) which can be critical is the application reviewed.

5.- From my point of view, the paper is unbalanced in the sense that the authors put a lot of effort in describing the methodology used, but lack technical/scientific descriptions and developments.

I hope my comments can help to enhance the quality of the manuscript.

Author Response

1.- References should be cited as [1-3] instead of [1,2,3]. Your reference manager should allow this format.

Thank you, we have revised this throughout.

2.- The Introduction section must be redone or complemented. It must include the purpose, importance and need of this study and the novelties and contributions with respect the state of the art.

The purpose and importance of a scoping review is to identify gaps and synthesize the research done in a specific area, as we have explained in the introduction and methods sections.  We have also explained that there are no scoping reviews to our knowledge that cover the same content as this paper, which is this paper’s novel contribution to this area.

3.- The authors devote a lot of efforts in explaining the screening methodology used but finally only 21 papers were selected for the core of the scooping review. Review paper usually are based on over 100 papers/documents.

The review is structured according to the PRISMA-SCR and Joanna Briggs Institute guidelines, citations are provided for this in the protocol section, so the methods section is predetermined in terms of content.  By definition, a scoping review synthesizes all papers found as a result of the exhaustive, structured search, and there is no minimum number for this to be valid.  In addition, we cite eight review papers in the manuscript which review significantly less than 100 papers and have been published in peer-reviewed journals.  When considering the Sensors journal in particular, the five most recently published scoping reviews consider 41 (https://www.mdpi.com/1424-8220/23/18/7686), 23 (https://www.mdpi.com/1424-8220/23/12/5732), 23 (https://www.mdpi.com/1424-8220/23/11/5048), 15 (https://www.mdpi.com/1424-8220/23/9/4530) and 9 (https://www.mdpi.com/1424-8220/23/7/3597) papers respectively.

4.- From my point of view this study misses important points such as a scientific discussion of the minimum video frame rate according to each specific application (speed race, long jump,…), positioning accuracy (the positioning accuracy of GPS or Wi-Fi systems, typically some few meters) which can be critical is the application reviewed.

This review specifically focuses on using smartphone accelerometers to measure walking gait kinematics.  When video is used as a comparator, we have included the capture rate, but this is not used in any of the apps.  As walking is used as the modality, “speed race” and “long jump” are also outside of the scope of this review.  Similarly, GPS and wifi are not relevant, as these do not feature in any of the apps, which all use the smartphone internal accelerometer sensors.

5.- From my point of view, the paper is unbalanced in the sense that the authors put a lot of effort in describing the methodology used, but lack technical/scientific descriptions and developments.

As explained above, this review follows specific reporting guidelines which dictate the content included.  Descriptions and synthesis in terms of comparator and apps considered has been included, as has a description of the methods to determine validity and/or reliability in each paper.

Round 2

Reviewer 2 Report

The authors have replied my comments